# Evaluation of Platelet-Derived Extracellular Vesicles in Gingival Fibroblasts and Keratinocytes for Periodontal Applications

**DOI:** 10.3390/ijms23147668

**Published:** 2022-07-11

**Authors:** Miquel Antich-Rosselló, Marta Munar-Bestard, Maria Antònia Forteza-Genestra, Javier Calvo, Antoni Gayà, Marta Monjo, Joana M. Ramis

**Affiliations:** 1Cell Therapy and Tissue Engineering Group, Research Institute on Health Sciences (IUNICS), University of the Balearic Islands (UIB), Ctra. Valldemossa km 7.5, 07122 Palma, Spain; miquel.antich1@estudiant.uib.es (M.A.-R.); marta.munar@uib.es (M.M.-B.); maria.forteza@ssib.es (M.A.F.-G.); jcalvo@fbstib.org (J.C.); agaya@fbstib.org (A.G.); 2Health Research Institute of the Balearic Islands (IdISBa), Carretera de Valldemossa, 79, 07120 Palma, Spain; 3Fundació Banc de Sang i Teixits de les Illes Balears (FBSTIB), Carrer de Rosselló i Cazador, 20, 07004 Palma, Spain; 4Departament de Biologia Fonamental i Ciències de la Salut, University of the Balearic Islands (UIB), Ctra. Valldemossa km 7.5, 07122 Palma, Spain

**Keywords:** extracellular vesicles, platelet lysate, in vitro wound healing, hyaluronic acid, regeneration, gingival fibroblasts, gingival keratinocytes

## Abstract

Gingival regeneration aims at restoring the architecture and functionality of oral damaged tissue. Different biomaterials or biological materials have been tested for tissue repair, such as platelet concentrates such as PL. In this article, the use of extracellular vesicles (EVs) derived from platelet lysate (PL) and their combination with hyaluronic acid biomaterials (HA) in an in vitro wound healing assay is investigated. EVs were isolated by size exclusion chromatography from PL. In addition, HA gels were formulated with PL or EVs. EVs or HA combined with EVs (HA-EVs) were tested in vitro in gingival fibroblasts and keratinocytes for biocompatibility (LDH activity and metabolic activity) and by an in vitro wound-healing assay and gene expression analysis. EVs and EVs-HA treatments were biocompatible in gingival fibroblasts and keratinocytes and showed an increase in wound healing in vitro compared to control. Moreover, changes in gene expression related to extracellular matrix remodeling were observed after the treatment with EVs. EVs can be combined with HA biomaterials, showing good biocompatibility and preserving their activity and functionality. Therefore, platelet-derived EVs could emerge as a new application for periodontal regeneration in combination with biomaterials in order to enhance their clinical use.

## 1. Introduction

Oral wound healing is a necessary process to form a biological barrier in the gingival tissues to avoid infections. In fact, periodontal sealing is especially relevant after surgical interventions or dental implantations [1]. Thus, current gingival regenerative approaches pursue a rapid wound-healing response and tissue regeneration among the different strategies under research, wherein the use of platelet concentrates, such as platelet-rich plasma (PRP) or platelet lysate (PL), has emerged due to platelet major role during wound healing [2]. Research has shown that platelet concentrates can ease the regeneration of dental or periodontal tissues [3]. Moreover, some studies show that platelets induce a wound-healing response in fibroblasts and keratinocytes [4,5,6]. Although the wound-healing response induced by platelets has traditionally been associated to the direct effects of secreted biomolecules such as growth factors, recently it has been suggested that extracellular vesicles (EVs) could also be involved in the wound-healing response [7,8].

EVs are membranous particles, naturally released by cells, which play a key role in cell communication [9]. They present a wide range of sizes, which extend from 30 nm up to 1000 nm. EVs have been classified according to their origin. Thus, exosomes are derived from the endosomal system, while microvesicles are generated by plasmatic membrane budding [10,11]. EVs are enriched in protein, lipids, and nucleic acids (DNA, mRNA, miRNA, tRNA), which allows for communication with the receptor cell [12]. Moreover, the lipid bilayer of EVs protects the biomolecules they contain, preventing their degradation once released to the extracellular space [9]. Their specific cargo depends on the cellular origin and conditions, with immune cells, mesenchymal stem cells, body fluids, and dental related cells being the most studied for oral regenerative medicine [8]. However, only few studies have explored the role of platelet-derived EVs on oral wound healing, although EVs have already been suggested as major effectors of platelet concentrate activity [13].

Moreover, the combination of biomaterials with EVs may improve tissue regeneration capability [14]. Among the different biomaterials that exist, hyaluronic acid (HA) biomaterials have been widely studied in periodontal regeneration [15,16]. HA is a component of the extracellular matrix able to modulate the regenerative process [17]. Therefore, HA-based biomaterials combined with platelet EVs could be good candidates for oral regenerative treatments.

Although platelet-derived EVs have been proposed as migration and wound healing promoters in dermal fibroblasts and keratinocytes [18,19], lack of evidence remains in the use of platelet EVs in gingival tissue. Gingival fibroblasts are the major constituents of periodontal connective tissue. They maintain gingival tissue integrity by producing and organizing different extracellular matrix components such as collagen and proteoglycans [20]. On the other hand, gingival keratinocytes form the outer layer of gingival tissue and create a protective barrier against external agents [21]. After a wound, gingival keratinocytes rapidly move to the injured area in order to cover the exposed connective tissue [20]. Thus, both gingival fibroblasts’ and keratinocytes’ migration response requires investigation when evaluating any regenerative agent. However, the factors that affect the regeneration of periodontal tissue are complex, and the direct equivalence between the fibroblasts or keratinocytes cell culture to in vivo applications is limited. Further studies evaluating protein levels and using more complex wound healing models, such as ex-vivo gingiva explants or three-dimensional oral mucosa models or in vivo animal models should be performed.

In the present study, we evaluated the effect of PL-derived EVs on gingival keratinocyte and fibroblast cell migration and gene expression after wounding, both the PL-derived EVs alone, as well as combined with HA gels for periodontal applications.

## 2. Results

### 2.1. Platelet-Derived EV Characterization

EV isolation and characterization were performed according to ISEV recommendations (Figure 1) [22]. Wide-field TEM images were used to confirm the presence of vesicles bodies (Figure 1A). Then, EVs were analyzed by NTA, showing a heterogeneous population with a median size ± standard error of 106.0 ± 2.5 nm (Figure 1B). Moreover, through the particle content (3.8·1011 particles/mL) and the protein concentration (167 µg/mL), the purity ratio of 2.3·109 particles/µg was obtained [23]. Specific EV markers (CD9 and CD63) were confirmed by Western blot (Figure 1C).

### 2.2. PL and EVs’ Effect on ihGK and ihGF

To evaluate the effects of PL and the PL-derived EVs, an in vitro functional study was performed using ihGK and ihGF (Figure 2). The wound closure area, metabolic activity, and LDH activity were determined 3 h after the wound for ihGK (Figure 2A–D) and 24 h after the wound for ihGF (Figure 2E–H). In both cell lines, EVs presented a statistically significant increase in wound closure compared to the control. In addition, in ihGF, wound closure was also higher on the EV-treated group compared to the PL-treated group (Figure 2A,B,E,F). No differences were observed for metabolic activity at 3 h in ihGK, but EV-treated fibroblasts showed higher metabolic activity than the control 24 h after wound and treatment (Figure 2C,G). No cytotoxicity was found for any of the groups (Figure 2D,H).

Additionally, mRNA expression levels for different marker genes were evaluated for both cell lines (Figure 3). PL- and EV-treated ihGK for 3 h showed a significant increase in *FN1* mRNA expression, but no differences were observed for *VIM* mRNA levels (Figure 3A). *COL1A1*, *DCN*, *MMP1*, *TIMP-1*, *ACTA2*, *TGF-β1*, and *EDN* were also tested for ihGK but not detected. However, ihGF treated with EVs presented increased levels for all the genes except for *DCN*, reaching significance for *COL1A1*, compared to control and PL, and *MMP1* and *EDN* compared to control (Figure 3B,C).

### 2.3. HA Gel Characterization

The equilibrium swelling ratio was determined for the different HA gels (Figure 4A). As expected, higher ESR were observed for all gel formulations at 24 h than at 3 h. However, HA-PL and HA-EVs presented lower ratios than HA in both time points, reaching statistical significance for the ESR for HA EVs at 3 h compared to the control. Additionally, the EV release over time for the HA-EVs group was determined by NTA (Figure 4B). HA-EVs presented a sustained release over the time period evaluated, with a higher amount of particle released at 24 h than at 3 h (Figure 4B).

### 2.4. In Vitro Effect of the Different HA Gels

The different HA gels were evaluated in an in vitro functional study (Figure 5). For ihGK, the cell closure area was significantly higher for HA-PL and HA-EVs after 3 h of treatment compared to the control (Figure 5A,B). Moreover, an increase in metabolic activity was observed for HA-EVs in comparison to the control (Figure 5C). No cytotoxicity was induced by any of the treatment groups in ihGK cells (Figure 5D). In addition, ihGF treated with HA, HA-PL, or HA-EVs also showed an increased wound closure area compared to the control. Moreover, HA-EV-treated cells presented a significantly more closed wound compared to the HA treated group (Figure 5E,F). Higher metabolic activity in ihGF was observed for all treated groups compared to the control one (Figure 5G). Furthermore, none of the treatments induced any cytotoxicity as measured by LDH activity released to the cell culture media (Figure 5H).

Gene expression was determined for cells treated with the different HA-formulated gels (Figure 6). ihGK treated with HA-EVs presented no significant differences in mRNA expression levels of VIM nor FN1 (Figure 6A). *COL1A1*, *DCN*, *MMP1*, *TIMP-1*, *ACTA2*, *TGF-β1*, and *EDN* were also tested for ihGK but not detected. Furthermore, ihGF treated with HA-EVs showed higher expression levels of *COL1A1* mRNA, compared to all other groups; *MMP1* mRNA, compared to the C and HA groups, and *TIMP-1* mRNA compared to HA. Moreover, HA-PL and HA also presented significantly higher levels of *MMP1* mRNA compared to the control. In addition, *DCN* mRNA levels of HA-EVs decreased when compared to HA (Figure 6B). Finally, HA-EVs showed decreased expression levels of *ACTA2* and an increase in *EDN* mRNA compared to the control and HA groups and *TGF-β1* mRNA of HA-EVs compared to HA, while HA-PL also reached a significant decrement in *ACTA2* mRNA levels and an increase on *EDN* mRNA compared to the control (Figure 6C).

## 3. Discussion

Gingival epithelial wound healing and regeneration is essential for tooth and periodontal tissue protection. In the last decade, platelet concentrates have increasingly been used for dental and periodontal tissue regeneration [2,3]. Here, we propose the use of PL-derived EVs for gingival wound closure, proving their in vitro effects on gingival keratinocytes and fibroblasts, both on its own and combined with hyaluronic acid gels.

In this study, we demonstrate that PL-derived EVs increase wound closure both in gingival keratinocytes and fibroblasts, despite being aware that cell lines may not behave like primary cells [24]. Previous studies had already proven the regenerative effect of platelets, but these effects were mainly attributed to the released proteins, such as growth factors [4,5,6]. Our results show higher effects for the groups using highly purified EVs, suggesting EVs as improved healing effectors of PL activity. As we and others already suggested [25,26], our results obtained with PL-derived EVs versus PL could be explained by a protective encapsulation of active biomolecules offered by the vesicles themselves. Therefore, in the future, platelet-derived EVs could be an effective substitute for platelet concentrates in regenerative medicine. In fact, further research could lead to an allogenic and an off-the-shelf therapy that would benefit the field [27]. A highlight of this work is the high purity of the used EVs since a purification by size exclusion chromatography was performed and only the fractions rich on EVs and poor on protein content were used. Isolated EVs were characterized according to the ISEV recommendations [22], presenting an heterogeneous population of small EVs positive in CD9 and CD63.

Additionally, in this study, PL and the EVs have been combined with HA gels. The HA gel formulation enables a topical or intradermal application that would allow a future clinical use and industrial development for oral interventions [28]. HA is a highly porous and hydrophilic biomaterial that allows for a proper interaction with the physiological environment [29]. The ESR values obtained are relatively high, but they decrease when HA is combined with PL or EVs, probably due to their lipophilic nature. Our in vitro studies show that wound healing assay response was improved in both gingival cell types by HA treatment as expected from previous reports [28]. What is more, this effect was enhanced by the combination of HA with PL or EVs, showing HA-EVs as having the best result in agreement with the effects observed for non-formulated EVs.

After a wound, a series of sequential responses are activated to reconstitute the damaged tissues. On one hand, diverse cell types must migrate, proliferate, and differentiate; on the other hand, these cells must secrete and organize several molecules of the extracellular matrix, including collagens, fibronectin, and proteoglycans [30]. These processes are governed by a complex array of signals and involve the regulation of the expression of numerous genes [20]. Here, we evaluated gene expression of different markers that are involved in these steps on the basis of previous studies [31]. Direct use of EVs or PL induced an increase in *FN1* mRNA levels after the scratch on ihGK, despite not reaching the twofold increase. Nevertheless, HA combined treatments do not present such increase, probably because HA is an extracellular matrix glycosaminoglycan involved in cell matrix formation [32], thus avoiding its expression when HA is supplied to the cells. Similar effects are observed towards *COL1A1* and *DCN* expression in HA-treated ihGF, further indicating the direct functionality associated to the hyaluronic acid itself. In addition, when EVs were used as treatment for their own or combined with HA, a 2.24-fold increase on *COL1A1* and a 2.17-fold increase on *MMP1* mRNA levels was observed, while when EVs were combined with HA, a 1.87-fold increase on *COL1A1* and a 4.45-fold increase on *MMP1* mRNA levels were observed. This suggests enhanced extracellular matrix remodeling, which has been suggested to promote a scarless and not fibrotic repair [33].

Moreover, EV-treated groups presented higher *EDN* mRNA levels in spite of not reaching the twofold increment, while when combined with HA, a 4.27-fold increase in *EDN* mRNA levels and a 0.69-fold decrease in *ACTA2* mRNA levels were observed. These results may indicate that HA-EV treatment would reduce the risk of fibrous tissue formation while maintaining the regenerative properties of EVs [34]. However, additional studies evaluating protein levels and using more complex wound healing models, such as ex vivo gingiva explants, three-dimensional oral mucosa models, or in vivo animal models, should be performed. As regards the biocompatibility, HA combined treatments presented low levels of LDH activity and improved levels of metabolic activity. Thus, according to ISO-10993:5, they are below the maximum allowed cytotoxic value for medical devices, indicating the safety of the treatment.

Overall, PL-derived EVs achieve improved regenerative effects compared to the direct use of PL. These effects may be caused by different properties of EVs. On the one hand, despite having normalized the protein amount, EVs may contain other functional biomolecules such as nucleic acids or active lipids [35]. For instance, platelet concentrates contain highly functional proteins that interact with the lipid metabolic pathways, which are highly relevant for tissue engineering applications [36]. On the other hand, EVs protect the cargo from degradation and enhance their delivery to the target cells [37]. Therefore, PL seems to present an increment of wound closure, probably caused mainly by proteins, while EVs have higher effects thanks to the encapsulation of these proteins and other active biomolecules and an enhanced delivery [38,39]. In fact, different miRNAs contained in EVs derived from different cell types have been suggested to modulate wound healing assay [40,41]. Nevertheless, further research should be performed to confirm the presence of these biomolecules in PL-derived EVs and their mechanism of action.

## 4. Materials and Methods

### 4.1. Human PL Preparation

Human fresh buffy coats were obtained from the IdISBa Biobank and their use for the current project was approved by its Ethics Committee (IB 1995/12 BIO). Blood donation criteria were met, excluding donors who had taken non-steroidal anti-inflammatory drugs. Platelet concentrates were obtained as previously described [25]. Briefly, six buffy coats were pulled, and a platelet concentrate was obtained after a centrifugation at 650× *g* for 10 min and leucocyte filtration. Then, at least three freeze/thaw cycles at −80 °C and 37 °C were performed to lyse platelets. Cell debris were discarded by centrifugation at 5050× *g* during 20 min at room temperature, and supernatant was filtered by 40.0 µm pore size membrane (Sartorius, Goettingen, Germany). PL was stored at −20 °C until use.

### 4.2. EVs Isolation

PL was centrifuged at 1500× *g* for 15 min at 4 °C. The supernatant was filtered through 0.8 µm porous membrane (Sartorius) for large cell debris elimination and then through 0.2 µm porous membrane (Sartorius). PL was centrifuged at 10,000× *g* for 30 min at 4 °C to preferentially retain small EVs. The supernatant (5 mL) was loaded on a Sepharose CL−2B precast column (GE Healthcare, Pittsburg, PA, USA). AKTA purifier system coupled with a collector Frac 950 (GE Healthcare) was used to set a flow rate at 0.5 mL/min. EVs were eluted with PBS (Capricorn, Ebsdorfergrund, Germany) in 5 mL fractions, which were collected and characterized (Appendix A). The ninth fraction was used for the experiments since it presented strong bands for the specific CD9 Western blot marker while preserving low levels of protein concentration.

### 4.3. Transmission Electron Microscopy (TEM)

Platelet-derived EVs were fixed in 2% formaldehyde (Sigma-Aldrich, Saint Louis, MI, USA) solution. Fixed EVs were set on copper formvar-carbon-coated grids (Ted Pella, Redding, CA, USA) during 20 min and washed with PBS. Then, the grids with the EVs on it were incubated with 1% glutaraldehyde (Sigma-Aldrich) for 5 min and washed with deionized water. The grids were stained for 1 min with 2% uranyl acetate (Electron Microscopy Sciences Hatfield, PA, USA) and washed with PBS. Images were taken using a TEM-H600 (Hitachi, Tokyo, Japan) at 50 kV.

### 4.4. Protein Quantification

Total protein amount of PL and EVs samples was quantified with BCA Protein Assay kit (Thermo Fisher, Waltham, MA, USA) following the manufacturer’s instructions. Absorbance was read at λ = 562 nm (BioTek Instruments, Winooski, VT, USA).

### 4.5. Western Blot

PL and EVs samples were prepared with non-reducing Laemli loading buffer and denatured at 70 °C. Samples were loaded in a 12% gradient SDS-PAGE gel, and proteins were separated by electrophoresis. The transfer was performed in humid conditions onto nitrocellulose membrane (GE Healthcare, Pittsburgh, PA, USA). A Ponceau S (Sigma-Aldrich) solution at 0.2% *v*/*v* was used for total protein visualization.

After several washes, membranes were blocked and incubated with anti-human CD9 (Thermo Fisher) and anti-human CD63 (Abcam, Cambridge, UK) antibodies. Secondary antibody incubation was performed with HRP-coupled anti-mouse IgG (Thermo Fisher). Chemiluminescence was induced with Clarity Western ECL Substrate (Bio-Rad, Hercules, CA, USA) and visualized after exposure on autoradiographic films (GE Healthcare).

### 4.6. Nanoparticle Tracking Analysis (NTA)

Size distribution and particle concentration were analyzed with Nanosight NS300 (Malvern Instruments, Malvern, UK). Samples were diluted (1:100) to a final volume of 1 mL and recorded with a laser at λ = 532 nm and an sCMOS camera. Data were analyzed with NTA 3.2 Dev Build 3.2.16 Software. Once the number of particles was set, purity ratio was calculated by the formula described by Webber et al. (Equation (1)) [23]:(1)$$\mathrm{Purity}\;\left(\mathrm{particles}/µg\right)=\frac{\mathrm{Particle}\;\mathrm{concentration}\;}{\mathrm{Protein}\;\mathrm{concentration}}$$

### 4.7. Hydrogel Preparations

Hydrogels based on hyaluronic acid (Bioibérica, F002103, *M*_w_ 800–1200 kDa, Spain) were prepared at 2% (*w*/*v*), according to its viscosity, swelling behavior, and the pore size of its microstructure by incubating overnight at 25 °C as previously described in a comparison study of different HA concentrations [42]. Thus, three different hydrogels were obtained: (a) hyaluronic acid hydrogel (HA) for which PBS was used; (b) PL containing hyaluronic acid hydrogel (HA-PL) for which PBS containing PL was used to obtain a final protein concentration of 0.167 µg/µL; (c) EVs containing hyaluronic acid hydrogel (HA-EVs), for which PBS containing EVs was used to obtain a final protein concentration of 0.167 µg/µL.

### 4.8. Equilibrium Swelling Ratio Determination

The equilibrium swelling ratio (ESR) of the hydrogels was determined for HA, HA-PL, and HA-EVs. A total of 1 mL of each gel was incubated in PBS at 37 °C for 3 h or 24 h. Then, each gel was centrifuged at 16,000× *g* for 15 min, and the wet pellets were weighted (W_w_); supernatants were stored at −80 °C for release studies. Then, hydrogels pellets were frozen at −80 °C and lyophilized during 72 h. Dried products were weighted again (W_d_). The experiment was performed in triplicate as previously described [42]. ESR values were determined using the following equation (Equation (2)):(2)$$\mathrm{ESR}\;=\frac{W_{w}-W_{d}\;}{W_{d}}\;$$

### 4.9. EVs Release

The supernatants obtained after the 16,000× *g* centrifugation step of the ESR determination were stored at −80 °C. HA-EVs supernatants were analyzed by nanoparticle tracking analysis to determine the number of particles released after 3 h or 24 h incubation during the swelling experiments.

### 4.10. Cell Culture

Immortalized Human Gingival Keratinocytes (ihGK, Applied Biological Materials Inc., Richmond, BC, Canada) and Immortalized Human Gingival Fibroblasts-hTERT (ihGF, Applied Biological Materials Inc) were grown at 37 °C and 5% CO_2_ atmosphere. The culture medium was renewed twice per week. Keratinocytes were cultured on tissue culture flasks for sensitive adherent cells (Sarstedt, Germany) using Dulbecco’s modified Eagle’s medium (DMEM) without magnesium and calcium (Gibco, Grand Island, NY, USA) and Ham’s F12 (Biowest, Nuaille, France) in a 2:3 proportion, supplemented with 0.01 mg/mL insulin (Sigma-Aldrich), 0.4 ng/mL hydrocortisone (Sigma-Aldrich), 6.7 ng/mL selenium (Sigma-Aldrich), 0.01 μg/mL human epidermal growth factor (ThermoFisher Scientific, Waltham MA, USA), 1M HEPES buffer (Biowest), 5.5 μg/mL transferrin (Sigma-Aldrich), 0.1 nM cholera toxin (Sigma-Aldrich), 2 mM L-glutamine (Sigma-Aldrich), 5% (*v*/*v*) fetal bovine serum embryonic stem cells tested (FBS, Biowest), and 100 µg/mL penicillin and 100 µg/mL streptomycin (Biowest). Fibroblasts were cultured in DMEM low glucose (Biowest) and Ham’s F12 (Biowest) in a 2:1 proportion, supplemented with 10% (*v*/*v*) FBS (Biowest) and 100 µg/mL penicillin, as well as 100 µg/mL streptomycin (Biowest).

### 4.11. Wound Healing In Vitro Assay

ihGK or ihFK were seeded in 48-well plates at a density of 20,000 cells/well. When confluence was reached, cells were washed twice with PBS and the medium was replaced by medium without supplements and containing 1% (*v*/*v*) EV’s depleted FBS. EV’s depletion was performed by ultracentrifugation at 120,000× *g* for 18 h at 4 °C. Three independent biological experiments were performed, having triplicates in each (*n* = 9). All experiments were conducted in parallel and under the same conditions. Thus, PL and EVs in vitro studies, either alone or combined with HA, present the same control group to compare the different effects.

Wound was performed by scraping the cell monolayer with a 100 µl sterile pipette tip in a straight line to create a scratch, cell medium was renewed, and the treatments were applied according to the group. The different groups tested were the control (having the medium renewed but no treatment), PL (5 μg of PL per well), EVs (5 μg of EVs per well), HA (30 μL of HA gel per well), HA-PL (30 μL of HA-PL gel per well, which contains 5 μg of PL), and HA-EVs (30 μL of HA-EVs gel per well, which contains 5 μg of EVs). The 5 µg dose was selected in agreement to previous experiments [25] and according to the dose–response evaluation of PL and EVs (Appendix A).

Images of the same areas were taken using a bright-field inverted microscope (Nikon Eclipse TS100) before treatment and 3 h after healing in ihGK and 24 h after healing in ihGF. Other time points were evaluated for both cell lines, but they proved to be too early or too late for wound analysis (Appendix A). The images were analyzed with ImageJ software v1.51k (NIH, Bethesda, MD, USA). The wound closure area (%) was defined as the difference of the scratch area before (A_i_) and after the treatment (A_f_) and normalized by the initial scratch area (Equation (3)).
(3)$$\mathrm{Wound}\;\mathrm{closure}\;\mathrm{area}\;\left(\%\right)=\frac{A_{i}-A_{f}\;}{A_{i}}\cdot 100\;$$

### 4.12. Cell Cytotoxicity

Cell media was collected after 3 h of treatment for ihGK and 24 h of treatment for ihGF. Lactate dehydrogenase (LDH) activity was measured with a Cytotoxicity Detection kit (Roche Diagnostics, Manheim, Germany) following the manufacturer’s instructions. For cytotoxicity calculation (Equation (4)), 0.1% Triton-X100-treated wells were used as high control, 100% cell death, while non-treated wells were used as low control, 0% cell death.
(4)$$\mathrm{Citotoxicity}\;\left(\%\right)=\frac{\mathrm{experimental}\;\mathrm{value}-\mathrm{low}\;\mathrm{control}\;}{\mathrm{high}\;\mathrm{control}-\mathrm{low}\;\mathrm{control}}\cdot 100\;$$

### 4.13. Metabolic Activity

Total metabolic activity was evaluated after performing the wound healing in vitro assays, that is, 3 h of treatment for ihGK and 24 h of treatment for ihGF. Presto Blue reagent (Life Technologies, Carlsbad, CA, USA) was used during 1 h of reagent incubation time following manufacturer’s protocol. Non-treated cells were set as 100%.

### 4.14. Gene Expression by Real-Time RT-PCR

RNA was isolated using RNAzol^®^ RT (Molecular Research Center, Cincinnati, OH, USA) from the cell culture monolayer after performing the wound healing in vitro assay and evaluating metabolic activity of the cells, 4 h for ihGK and 25 h for ihGF. RNA concentration was quantified with a NanoDrop spectrophotometer (NanoDrop Technologies, Wilmington, DE, USA) and normalized for reverse transcription to cDNA using a High Capacity RNA-to-cDNA kit (Applied Biosystems, Foster City, CA, USA).

Real-time PCR was performed for three reference genes and different target genes on the basis of previous studies [43] (Table 1) using the Lightcycler 480 thermocycler (Roche Diagnostics) and SYBR green detection. Each reaction well contained a Lightcycler 480 SYBR Green I Master (Roche Diagnostics), 0.5 μM of each primer, the sense and the antisense, and 3 μL of the cDNA dilution in a final volume of 10 μL. The amplification program started with a 5 min pre-incubation step for cDNA template denaturation at 95 °C, followed by 45 cycles consisting of 10 s steps of denaturation at 95 °C, annealing at 60 °C, and an extension at 72 °C. Fluorescence was measured at 72 °C after each cycle.

For relative quantification, standard curves were constructed for all genes. The Second Derivative Maximum Method provided by the LightCycler480^®^ analysis software version 1.5 (Roche Diagnostics) was used to calculate the amount of each gene from the crossing point data. Reference genes were used to normalize the target genes expression levels and changes were compared to control group, which was set to 100%.

### 4.15. Statistical Analysis

SPSS program version 25.0 (SPSS Inc., Chicago, IL, USA) was used for statistical comparison, and GraphPad Prism (version 7, La Jolla, CA, USA) was used to represent the data as mean values of the independent experiments ± SEM. In each independent experiment, at least two cell wells per group were evaluated; for the statistical analysis, we obtained the mean of the different wells of the same group and were considered as a single biological sample. The different treatments within each independent experiment were paired and evaluated through paired t-test analysis. Results were considered statistically significant at *p* < 0.05.

## 5. Conclusions

In conclusion, platelet-derived EVs present a good biocompatibility in vitro and induce cell migration in response to wounding on both gingival keratinocytes and fibroblasts. Moreover, EVs induce changes on expression levels of genes associated with the remodeling process during wound healing. Finally, EVs can be combined with HA gels to produce more applicable treatments while preserving their regenerative effects.

## Figures and Tables

**Figure 1 ijms-23-07668-f001:**
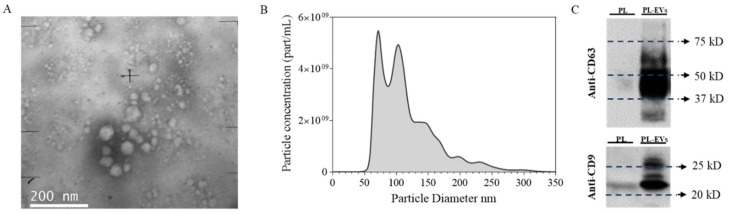
Platelet-derived extracellular vesicle characterization. (**A**) Wide-field TEM image of PL-EVs taken at ×100k augments. (**B**) Particle size distribution of EVs determined by NTA analysis. (**C**) Presence of EV biomarkers CD63 and CD9 for PL and PL-EVs. The same amount of protein was loaded per well (5 µg).

**Figure 2 ijms-23-07668-f002:**
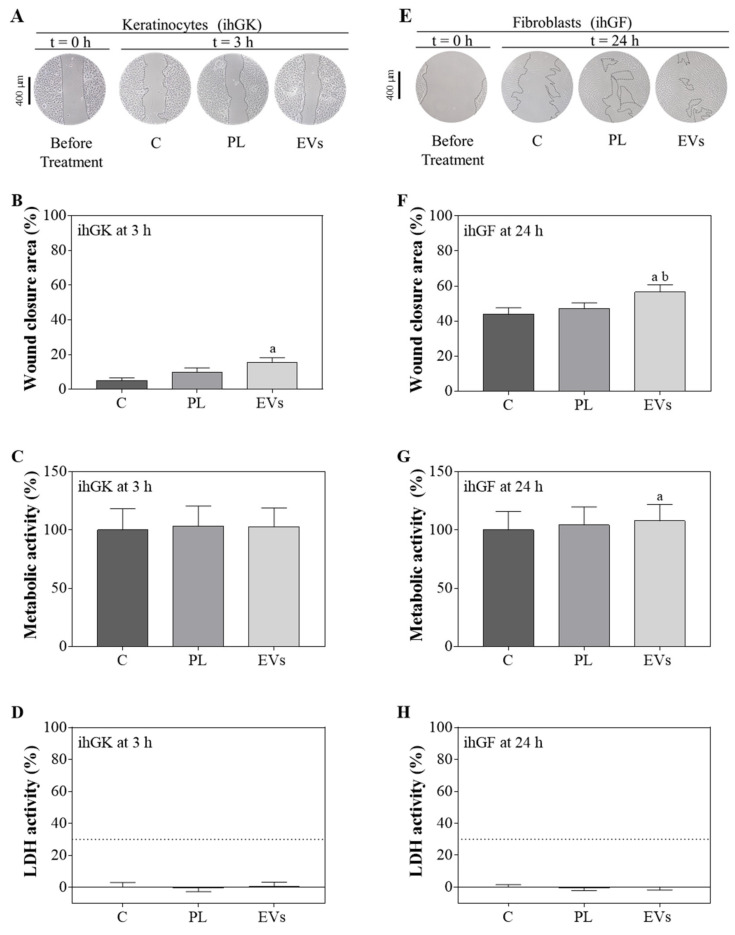
Effects of PL and EV treatments after the wound healing assay. (**A**) Images of ihGK cell morphology at t = 0 h and t = 3 h of treatment; images were taken at ×100 augments. (**B**) Wound closure area after 3h of treatment of ihGK. (**C**) Metabolic activity of ihGK after 3 h of treatment; data of the control group was set as 100%. (**D**) LDH activity measured in ihGK culture media after 3 h of treatment; control group was set as 0% of toxicity and culture media from cells treated with 1% Triton X-100 was set at 100%. A dashed line is shown at 30%, which is the maximum value accepted for cytotoxicity of medical devices according to ISO-10993:5. (**E**) Images of ihGF cell morphology at t = 0 h and t = 24 h of treatment; images were taken at a ×100 augments. (**F**) Wound closure area after 24 h of treatment of ihGF. (**G**) Metabolic activity of ihGF after 24 h of treatment; data of the control group was set as 100%. (**H**) LDH activity measured in ihGF culture media after 24 h of treatment; control group was set as 0% of toxicity, and culture media from cells treated with 1% Triton X-100 was set at 100%. A dashed line is shown at 30%, which is the maximum value accepted for cytotoxicity of medical devices according to ISO-10993:5. Values represent the mean ± SEM. For ihGK, 9 independent experiments, with at least duplicate wells per group (*n* = 9), were performed, while for ihGF, 10 independent experiments, with at least duplicate wells per group (*n* = 10), were performed. Results were statistically compared by paired *t*-test. Statistically significant differences were considered for *p* < 0.05 and represented with “a” compared to control or “b” compared to PL.

**Figure 3 ijms-23-07668-f003:**
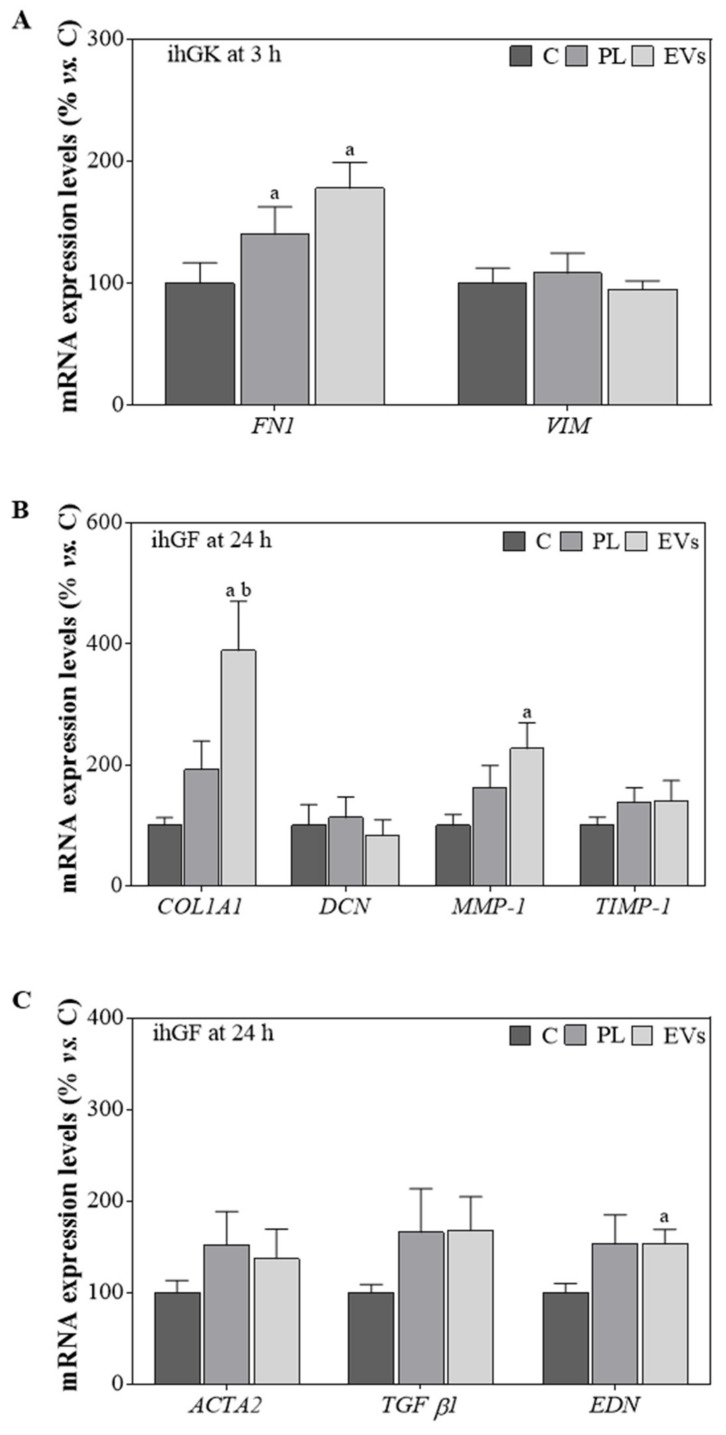
Gene expression levels in ihGK and ihGF after PL and EV treatment. (**A**) Effects on ihGK mRNA expression levels of *FN1* and *VIM* after 3 h of treatment. (**B**) Effects on ihGF mRNA expression levels of *COL1A1*, *DCN*, *MMP-1*, and *TIMP-1* after 24 h of treatment. (**C**) Effects on ihGF mRNA expression levels of *TGF-β1*, *EDN*, and *ACTA2* after 24 h of treatment. Values represent the mean ± SEM. For ihGK, 9 independent experiments, with at least duplicate wells per group (*n* = 9), were performed, while for ihGF, 8 independent experiments, with at least duplicate wells per group (*n* = 8), were performed. Results were statistically compared by paired *t*-test. Statistically significant differences were considered for *p* < 0.05 and represented with “a” compared to C and “b” compared to PL.

**Figure 4 ijms-23-07668-f004:**
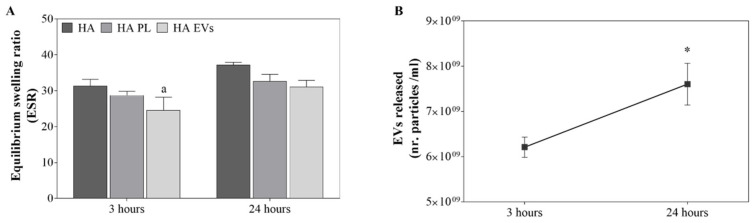
HA gel characterization. (**A**) Equilibrium swelling ratio (ESR) at 3 h and 24 h for each gel formulation. (**B**) EVs released per mL of HA-EVs gel at 3 h and 24 h. Values represent the mean ± SEM. Three independent samples (*n* = 3) were evaluated. ESR results were statistically compared by ANOVA using DMS as a post hoc, while EVs released were compared by independent samples t-test. Statistically significant differences were considered for *p* < 0.05 and represented with “a” compared to HA and * compared to 3 h.

**Figure 5 ijms-23-07668-f005:**
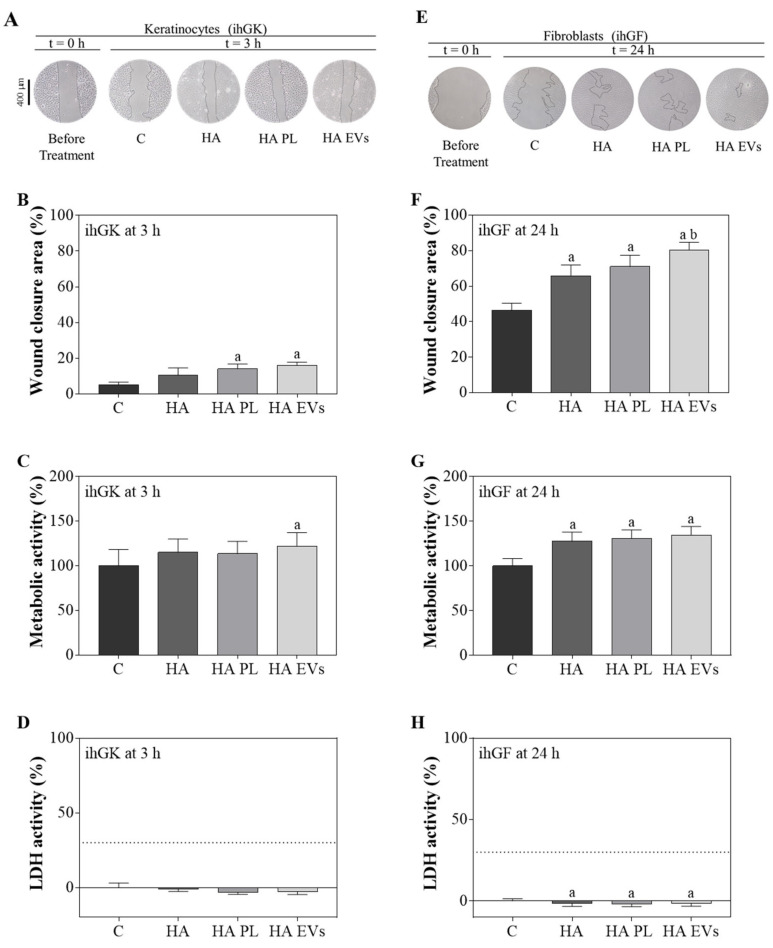
Effects of HA and its combination with PL and EV after the wound-healing assay. (**A**) Images of ihGK cell morphology at t = 0 h and t = 3 h of treatment; images were taken at a ×100 augments. (**B**) Wound closure area after 3 h of treatment of ihGK. (**C**) Metabolic activity of ihGK after 3 h of treatment; data of the control group was set as 100%. (**D**) LDH activity measured in ihGK culture media after 3 h of treatment; control group was set as 0% of toxicity and culture media from cells treated with 1% Triton X-100 was set at 100%. A dashed line is shown at 30%, which is the maximum value accepted for cytotoxicity of medical devices according to ISO-10993:5. (**E**) Images of ihGF cell morphology at t = 0 h and t = 24 h of treatment, images were taken at ×100 augments. (**F**) Wound closure area after 24 h of treatment of ihGF. (**G**) Metabolic activity of ihGF after 24 h of treatment, data of the control group was set as 100%. (**H**) LDH activity measured in ihGF culture media after 24 h of treatment; control group was set as 0% of toxicity and culture media from cells treated with 1% Triton X-100 was set at 100%. A dashed line is shown at 30%, which is the maximum value accepted for cytotoxicity of medical devices according to ISO-10993:5. Values represent the mean ± SEM. For ihGK, 9 independent experiments, with at least duplicate wells per group (*n* = 9), were performed, while for ihGF, 7 independent experiments, with at least duplicate wells per group (*n* = 7), were performed. Results were statistically compared by paired *t*-tests. Statistically significant differences were considered for *p* < 0.05 and represented with “a” compared to the control and “b” compared to HA.

**Figure 6 ijms-23-07668-f006:**
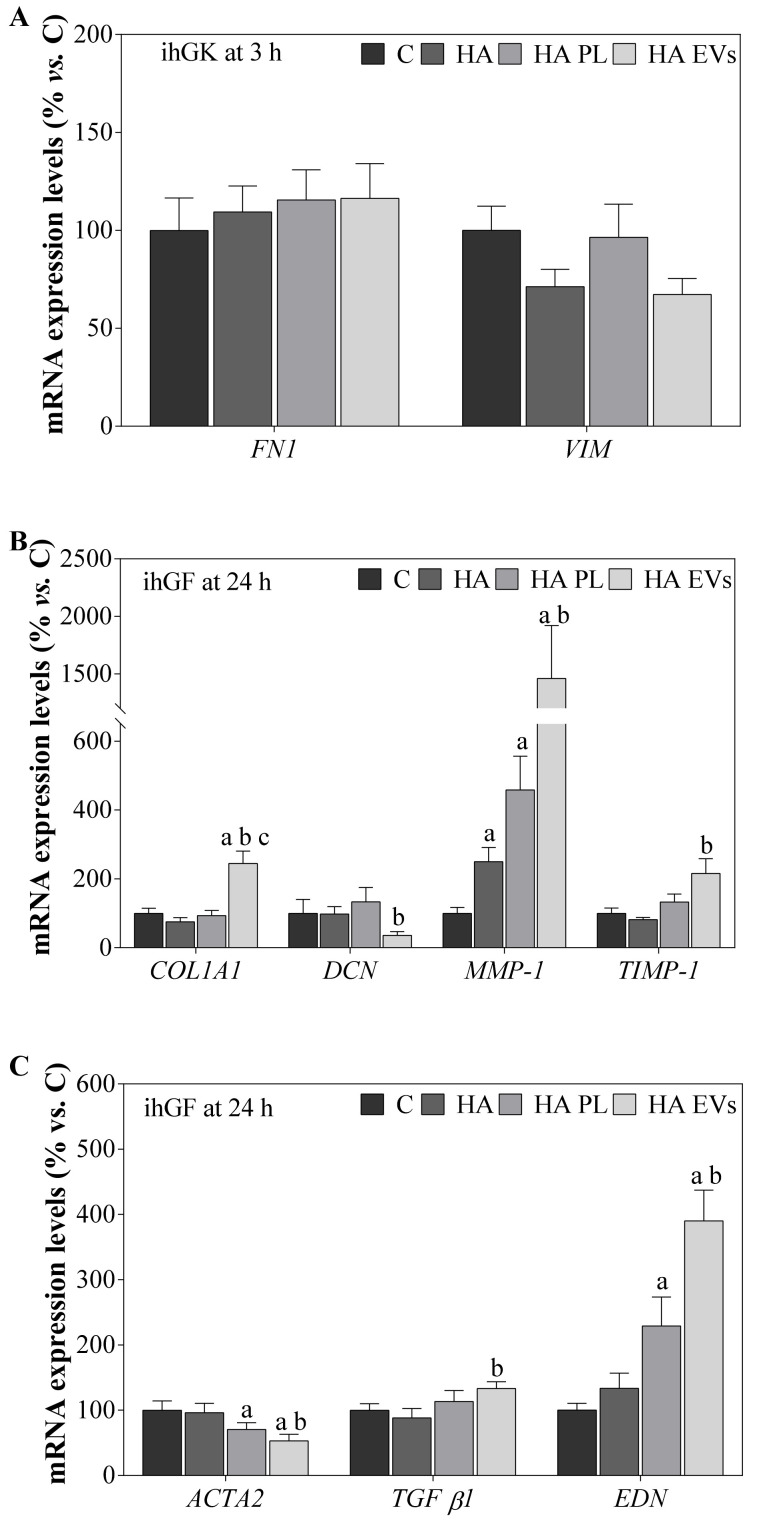
Gene expression levels in ihGK and ihGF after HA, HA-PL, and HA-EV treatment. (**A**) Effects on ihGK mRNA expression levels of *FN1* and *VIM* after 3 h of treatment. (**B**) Effects on ihGF mRNA expression levels of *COL1A1*, *DCN*, *MMP-1*, and *TIMP-1* after 24 h of treatment. (**C**) Effects on ihGF mRNA expression levels of *TGF-β1*, *EDN*, and *ACTA2* after 24 h of treatment. Values represent the mean ± SEM. For ihGK, 9 independent experiments, with at least duplicate wells per group (*n* = 9), were performed, while for ihGF, 7 independent experiments, with at least duplicate wells per group (*n* = 7), were performed. Results were statistically compared by paired *t*-test. Statistically significant differences were considered for *p* < 0.05 and represented with “a” compared to C, with “b” compared to PL and with “c” compared to HA PL.

**Table 1 ijms-23-07668-t001:** Genes and their sense (S) and antisense (A) primer sequences used in the real-time RT-PCR.

Gen	Primer Sequence (5′−3′)	Product Size (bp)	Function	Ref.
***Collagen I α1 (COL1A1)*** NM_000088.3	Sense: CCTGACGCACGGCCAAGAGG Antisense: GGCAGGGCTCGGGTTTCCAC	122	COL1A1 is one of the main components of the extracellular matrix.	[44]
***Decorin (DCN)>*** NM_001920.3	Sense: ATCTCAGCTTTGAGGGCTCC Antisense: GCCTCTCTGTTGAAACGGTC	146	DCN is a component of the extracellular matrix. However, *DCN is also* related to quiescence and growth inhibition [45].	[44,45]
***Matrix metalloproteinase-1 (MMP-1)*** NM_002421.3	Sense: TGTCAGGGGAGATCATCGGGAC Antisense: TGGCCGAGTTATGAGCTGCA	177	MMP1 is an enzyme that degrades collagen proteins.	[46]
***Tissue inhibitor of metalloproteinases 1 (TIMP-1)*** NM_003254.2	Sense: TTCCGACCTCGTCATCAGGG Antisense: TAGACGAACCGGATGTCAGC	144	TIMP-1 inhibits MMP-1 and it is involved in extracellular matrix remodeling.	[47]
***α-Smooth muscle actin 2 (ACTA2)*** NM_001141945.1	Sense: TAAGACGGGAATCCTGTGAAGC Antisense: TGTCCCATTCCCACCATCAC	184	It promotes collagen production, but it may induce myofibroblast differentiation and scar formation.	[34]
***Transforming growth factor-β1 (TGF-B1)*** NM_000660.4	Sense: TGTCACCGGAGTTGTGCGGC Antisense: GGCCGGTAGTGAACCCGTTG	131	It promotes COL1A1 production and enhances cell proliferation.	[34]
***Endothelin-1 (EDN)*** NM_001955.4	Sense: ACGGCGGGGAGAAACCCACT Antisense: ACGGAACAACGTGCTCGGGA	147	It promotes COL1A1 production and enhances cell proliferation.	[34]
***Fibronectin (FN1)*** NM_001365522.2	Sense: CGGAGAGACAGGAGGAAATAGCCCT Antisense: TTGCTGCTTGCGGGGCTGTC	150	Fibronectin is a component of the extracellular matrix that induces wound healing and cell adhesion, migration, and differentiation.	[48]
***Vimentin (VIM)*** NM_003380.5	Sense: GGCCGCCTGCAGGATGAGATTC Antisense: CAGAGAAATCCTGCTCTCCTCGC	153	VIM is associated with the cytoskeleton reorganization and induces cell proliferation and migration during the regeneration process in keratinocytes.	[49]
***Glyceraldehyde-3-phosphate dehydrogenase (GAPDH)*** NM_002046.3	Sense: TGCACCACCAACTGCTTAGC Antisense: AAGGGACTTCCTGTAACAA	87	Housekeeping gene.	
***Beta-actin (ACTBL2)*** NM_001101.3	Sense: CTGGAACGGTGAAGGTGACA Antisense: AAGGGACTTCCTGTAACAA	140	Housekeeping gene.	
***18S ribosomal RNA (18S rRNA)*** NR_146156.1	Sense: GTAACCCGTTGAACCCCATT Antisense: CCATCCAATCGGTAGTAGCG	151	Housekeeping gene.	

## Data Availability

The data present in this study are available on request from the corresponding authors.

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
