# Peer review of "Evaluation of Platelet-Derived Extracellular Vesicles in Gingival Fibroblasts and Keratinocytes for Periodontal Applications"

_ijms, 2022, doi:10.3390/ijms23147668_

Round 1
Reviewer 1 Report
The authors used an in vitro model based on fibroblasts/keratinocytes, to prove the hypothesis that extracellular vesicles from platelet lysate, in combination with other biomolecules, could contribute to faster healing of oral damaged tissue.
1. Factors affecting the periodontal tissue regeneration are complex and the direct equivalence between fibroblasts/keratinocytes metabolic rate would not suffice for a sound dissertation on in vivo oral damaged tissue healing. The reader would benefit from a little more detailed introduction in this sense, even if the subject was addressed in the discussion chapter. Nevertheless, the principle of the investigation is clear and this observation is not affecting the base idea of the manuscript.
2. For the effects of PL and EV treatments, in figure 2, authors afirm that "EV treated fibroblasts showed higher metabolic activity than 263 the Control 24h after wound and treatment" (lines 262-264). The claimed metabolic increase seem to be into the margins of the confidence interval. A detailed explanation of the finding is needed.
3. For the gene expression study, a discussion about the fold increase relevancy and gene related significance cutoffs may be a plus. A commonly accepted fold increase in gene expression of 2 is commonly seen as significant, while using different stringency criteria one could alter the initial interpretation. The dynamics of the gene expression changes are also important, so an argumentation of the 3 hours end-point analysis is important.
4. Lines 419-420 phrasing could be improved.
5. Citations would help the affirmation in lines 412-414.
The discussion is pertinent and successful to convince that HA-EV treatment in this in vitro model induces a level of modification in metabolic rates and gene expression patterns. Even if the translation of those results in vivo may or may not confirm the findings, the manuscript is an interesting collection of experiments and scientific inquiries.
Author Response
Response to reviewer 1:
The authors used an in vitro model based on fibroblasts/keratinocytes, to prove the hypothesis that extracellular vesicles from platelet lysate, in combination with other biomolecules, could contribute to faster healing of oral damaged tissue.
- Factors affecting the periodontal tissue regeneration are complex and the direct equivalence between fibroblasts/keratinocytes metabolic rate would not suffice for a sound dissertation on in vivo oral damaged tissue healing. The reader would benefit from a little more detailed introduction in this sense, even if the subject was addressed in the discussion chapter. Nevertheless, the principle of the investigation is clear and this observation is not affecting the base idea of the manuscript.
In agreement with the reviewer’s suggestion a brief explanation has been added to the introduction section in order to address the in vitro limitations.
- For the effects of PL and EV treatments, in figure 2, authors affirm that "EV treated fibroblasts showed higher metabolic activity than 263 the Control 24h after wound and treatment" (lines 262-264). The claimed metabolic increase seem to be into the margins of the confidence interval. A detailed explanation of the finding is needed.
For understanding the significant difference found and in order to have independent samples in the in vitro studies, the experiments were performed by different researchers (different operators performing the wounds), at different days and using cells from different passages. Then, as explained in the methods section, the graphs show mean values ± SEM, but, for the statistical analysis, for each independent experiment we obtained the mean of the different wells of the same group and were considered as a single biological sample. The different treatments within each independent experiment were paired and evaluated through paired t-test analysis. This means that in each experiment (performed in the same day using samples from the same passage and wounds performed by the same operator) a metabolic increase was observed when comparing control and the EV treated group, though variability among different experiments, when pulled together might mask these differences.
- For the gene expression study, a discussion about the fold increase relevancy and gene related significance cutoffs may be a plus. A commonly accepted fold increase in gene expression of 2 is commonly seen as significant, while using different stringency criteria one could alter the initial interpretation. The dynamics of the gene expression changes are also important, so an argumentation of the 3 hours end-point analysis is important.
In agreement with the reviewer’s comment the discussion has been modified adding the fold-changes in gene expression.
- Lines 419-420 phrasing could be improved.
Thank you for your comment, this sentence has been changed.
- Citations would help the affirmation in lines 412-414.
In agreement with the reviewer’s comment, a related study has been cited regarding the role of those genes in tissue regeneration.
The discussion is pertinent and successful to convince that HA-EV treatment in this in vitro model induces a level of modification in metabolic rates and gene expression patterns. Even if the translation of those results in vivo may or may not confirm the findings, the manuscript is an interesting collection of experiments and scientific inquiries.

Reviewer 2 Report
· Introduction. Line 34. "Platelet-Rich Plasma" is preferred to "Platelet Rich Plasma" (Mesh term in Pubmed: https://www.ncbi.nlm.nih.gov/mesh/68053657)
· Materials and Methods. Line 79. Typo, reference [22] before the point.
· Materials and Methods. Lines 82,82, etc. please correct the symbol of degree "ºC" and not "ºC"
· Materials and Methods. Lines 93-94. The authors should detail why only the ninth fraction was used.
· Materials and Methods. Transmission Electron Microscopy (TEM). Please, confirm that the samples were processed (gluta., uranyl,…) in the grid, and not previously to set in it.
· Materials and Methods. Equilibrium swelling ratio determination. Please, the authors should reference the method. I'm sorry, but I don't fully understand the method, since it is not a typical swelling test.
· Materials and Methods. Wound healing in vitro assay. Line 169: "Cells were seeded" Please, specify which cells (ihGK, ihGF)
· Materials and Methods. Wound healing in vitro assay. Line 189. ImageJ software. Please, cite properly ImageJ software…
· Materials and Methods. Wound healing in vitro assay. Please, the authors must check the Equation 3. Could be Ai the denominator?
· Results. 3.1. Platelet-derived EVs characterization. The authors state that "EV isolation and characterization were performed according to ISEV recommendations" citing [33]. Please, provide as supplementary material the ISEV 2018 checklist.
· Discussion. The authors state that "On the one hand, despite having normalized the protein amount, EVs may contain other functional biomolecules such as nucleic acids or active lipids". I believe that PRPs (including PL) have these components (in the plasma, and in the releaseate of the platelets) For example: https://pubmed.ncbi.nlm.nih.gov/23505226/ for lipids. Please, discuss this.
· Discussion. "On the other hand, EVs protect the cargo from degradation and enhance their delivery to the target cells [44]." I this that this good argument. Great.
· Discussion. This affirmation "Therefore, platelet-derived EVs could be an effective substitute for platelet concentrates in regenerative medicine"…I sincerely believe that this statement is a bit daring. I agree with the authors that EV therapy may be relevant in the future, but at present I see its clinical translation as complicated. The authors can discuss a bit about the clinical translation of platelet EVs, and in what context (perhaps allogeneic?, For example https://pubmed.ncbi.nlm.nih.gov/27908451/).
· In conclusion, a great work of the authors. !GRAN TRABAJO! I encourage them to continue in this line. Please, correct and complete the points.
Author Response
Response to reviewer 2:
Introduction. Line 34. "Platelet-Rich Plasma" is preferred to "Platelet Rich Plasma" (Mesh term in Pubmed: https://www.ncbi.nlm.nih.gov/mesh/68053657)
In agreement with the reviewer’s suggestion, “Platelet Rich Plasma” has been renamed to “Platelet-Rich Plasma”.
Materials and Methods. Line 79. Typo, reference [22] before the point.
Thank you for the observation, this typographic mistake has been amended.
Materials and Methods. Lines 82,82, etc. please correct the symbol of degree "ºC" and not "ºC"
In agreement with the reviewer’s observation the degree symbols have been provided without the underline.
Materials and Methods. Lines 93-94. The authors should detail why only the ninth fraction was used.
In agreement with the reviewer’s suggestion a brief explanation has been provided regarding the use of the 9th fraction for the experiments.
Materials and Methods. Transmission Electron Microscopy (TEM). Please, confirm that the samples were processed (gluta., uranyl,…) in the grid, and not previously to set in it.
In agreement with the reviewer’s observation a more detailed explanation has been provided for samples preparations for TEM to avoid confusion.
Materials and Methods. Equilibrium swelling ratio determination. Please, the authors should reference the method. I'm sorry, but I don't fully understand the method, since it is not a typical swelling test.
In agreement with the reviewer’s suggestion the method has been referenced. Usually, swelling tests dry the gels by heating them up, but since we have combined gels with biomolecules freeze dry processes allow to maintain the integrity of thegels and the molecules and avoid unexpected loses.
Materials and Methods. Wound healing in vitro assay. Line 169: "Cells were seeded" Please, specify which cells (ihGK, ihGF)
In agreement with the reviewer’s suggestion, the word “cells” has been replaced for “ihGK or ihGF” in order to state that both cells were used for the wound healing assay and avoid misunderstandings.
Materials and Methods. Wound healing in vitro assay. Line 189. ImageJ software. Please, cite properly ImageJ software…
We have properly cited the ImageJ software according to the reviewer’s suggestion.
Materials and Methods. Wound healing in vitro assay. Please, the authors must check the Equation 3. Could be Ai the denominator?
Thank you for your observation, the equation has been amended with Ai in the denominator.
Results. 3.1. Platelet-derived EVs characterization. The authors state that "EV isolation and characterization were performed according to ISEV recommendations" citing [33]. Please, provide as supplementary material the ISEV 2018 checklist.
The ISEV 2018 checklist needs to be contextualized with the whole publication, and we believe that we need a specific authorization to publish the entire list as part of our supplementary material. Moreover, the citation of the open access paper is provided, meaning that any reader has it available at any time.
Discussion. The authors state that "On the one hand, despite having normalized the protein amount, EVs may contain other functional biomolecules such as nucleic acids or active lipids". I believe that PRPs (including PL) have these components (in the plasma, and in the releaseate of the platelets) For example: https://pubmed.ncbi.nlm.nih.gov/23505226/ for lipids. Please, discuss this.
Thank you for your suggestion, we have added this part of discussion.
Discussion. "On the other hand, EVs protect the cargo from degradation and enhance their delivery to the target cells [44]." I this that this good argument. Great.
Thank you for your comment.
Discussion. This affirmation "Therefore, platelet-derived EVs could be an effective substitute for platelet concentrates in regenerative medicine"…I sincerely believe that this statement is a bit daring. I agree with the authors that EV therapy may be relevant in the future, but at present I see its clinical translation as complicated. The authors can discuss a bit about the clinical translation of platelet EVs, and in what context (perhaps allogeneic?, For example https://pubmed.ncbi.nlm.nih.gov/27908451/).
Thank you for your suggestion, we have added this part of discussion.
In conclusion, a great work of the authors. !GRAN TRABAJO! I encourage them to continue in this line. Please, correct and complete the points.
Thank you for your comment! We will try to continue in this line!
